# Cultural Adaptation and Psychometric Properties of the Trust Me Scale—Italian Version: A Validation Study

**DOI:** 10.3390/healthcare11081086

**Published:** 2023-04-11

**Authors:** Dhurata Ivziku, Rosario Caruso, Marzia Lommi, Gianluca Conte, Arianna Magon, Alessandro Stievano, Gennaro Rocco, Ippolito Notarnicola, Maddalena De Maria, Raffaella Gualandi, Daniela Tartaglini, Anna De Benedictis

**Affiliations:** 1Department of Health Professions, Fondazione Policlinico Universitario Campus Bio-Medico, 00128 Rome, Italy; 2Health Professions Research and Development Unit, IRCCS San Donato Hospital, San Donato Milanese, 20097 Milano, Italy; 3Department of Biomedical Sciences for Health, University of Milan, 20133 Milano, Italy; 4Unit Care to the Person, Local Healthcare Authority Rome 2, 00159 Roma, Italy; 5Department of Clinical and Experimental Medicine, University of Messina, 98100 Messina, Italy; 6Centre of Excellence for Nursing Scholarship, Order of Nurses of Rome, 00136 Rome, Italy; 7Degree Course in Nursing, Catholic University “Our Lady of Good Counsel”, 1000 Tirana, Albania; 8Department of Biomedicine and Prevention, University Tor Vergata, 00133 Rome, Italy; 9Vice President Italian Scientific Society for the Direction and Management of Nursing (SIDMI), 00198 Rome, Italy; 10Clinical Direction, Fondazione Policlinico Universitario Campus Bio-Medico, 00128 Rome, Italy

**Keywords:** trust, instrument, validation study, psychometric testing, validity, reliability, nurses, nurse manager, leadership

## Abstract

Background: The Trust Me Scale is a widely used instrument to measure trust in healthcare providers. However, no Italian version of the scale exists yet, limiting its use in Italian-speaking populations. The aim of this study is to translate and validate the Trust Me Scale for use in Italian-speaking populations in nurses and nurse managers. Methods: The translation process involved methodological steps of collaborative and iterative translation with cultural adaptation. The validation process included a cross-sectional study enrolling a convenience sample of 683 nurses and 188 nurse managers who completed the Italian version of the Trust Me Scale and measures of intention to leave, satisfaction, and organizational commitment. Results: Item 5 was removed for poor factor loading, and items 11 and 13 were removed following an a priori strategy focused on deleting items with correlations between residual variables different than expected based on theoretical expectations derived from previous research. The final model fit well to sample statistics with a three-factor structure (harmony, reliability, and concern) and 13 items. A multiple-indicator multiple-cause model showed a measurement invariance between nurses and nurse coordinators. Construct validity was also supported by the evidence that the measured domains of trust align with the theoretical expectations and are related to the intention to leave, job satisfaction, and organizational commitment. Each dimension showed adequate scale reliability. Conclusions: The Italian version of the Trust Me Scale is a valid and reliable instrument to measure trust in nurses and nurse managers in Italian-speaking contexts. It can be used for research in nursing and leadership and evaluation of interventions aimed at improving trust in healthcare contexts.

## 1. Introduction

The concept of trust has been widely studied across various disciplines, including sociology, psychology, philosophy, religion, and nursing [1]. It has been classified as a social construct [2], a psychological state [3], or an attitude or behavior [1]. Although it has been extensively analyzed, only in the last few years has there been an agreement regarding its multidimensional nature [1,3]. The dimensions of trust encompass a range of emotional constructs, such as confidence, respect, commitment, and teamwork, as well as cognitive constructs like understanding through knowledge, experience, and familiarity, and behavioral constructs, such as honesty, reliability, proactivity, performance, communication, and quality of interactions [2]. Despite the extensive research on the concept, there is currently no widely accepted and unique definition of trust [3,4]. In this research trust is defined as “*a willingness to increase one’s resource investment in another party, based on positive expectation, resulting from past positive mutual interactions*” [5]. This definition captures the three main elements (vulnerability, reciprocity, and expectation) of trust, as well as the cyclical nature of the trust process.

Trust is linked to the context: it is dynamic and fluctuates in intensity and significance over time [6]. It is crucial in fostering and enhancing favorable social interactions [3,6] and in supporting relationships between individuals (interpersonal) or between individuals and the organization (impersonal) [1]. Therefore, trust is an essential aspect of social exchange and plays a critical role in facilitating cooperation and collaboration between individuals and groups [7]. By understanding the referents of trust and how they impact social exchanges, researchers and organizations can create more positive and productive work environments that foster trust, cooperation, and success [7].

Trust is vital in creating a healthy organizational culture [5], particularly in complex and uncertain settings such as healthcare organizations [4]. In healthcare, trust is a critical element of the relationship between patients and healthcare professionals and among healthcare professionals themselves.

Trust has been studied in nursing [4] and leadership research [3] within healthcare organizations. In nursing research, trust has been examined from two perspectives: the nurse-patient relationship and relationships among healthcare professionals in the work context [1]. When the nurse-patient relationship is built on a foundation of trust, patients are more likely to feel comfortable, secure, and appreciate the compassion, respect, and understanding in their interactions with nurses, resulting in better outcomes and a more positive care experience [1]. From a work environment perspective, trust plays a critical role in facilitating the creation of efficient teams and fostering a positive relationship between staff and managers [1]. When team members trust each other, they are more likely to work collaboratively, share knowledge and information, and support one another to achieve common goals.

According to research in the field of leadership, nurse managers who adopt a supportive leadership style such as transformational [8,9], aesthetic [9], authentic, ethical, servant, and empowering [3] are likely to have a significant and positive impact on the level of trust that staff members have in their leader. Similarly, when staff and managers have a trusting relationship, it can create a supportive work environment where staff members feel valued, respected, and supported [4]. Moreover, trust is a crucial trait of effective leadership, and it is the responsibility of a good leader to foster trust within their team [10].

High levels of trust in work environments can result in various beneficial outcomes such as professionalism [4], effective decision-making [4,10], improved problem-solving [4], autonomy, adherence to professional values [8], proactive behavior [9], teamwork, support, cooperation, delegation [4], innovation [11], efficiency [4], reduced errors [10], reduced costs [2], and enhanced quality of care [4]. Conversely, low levels of trust in work environments can result in higher levels of stress, increased conflicts, absenteeism, turnover, and lower standards of care [4]. Regarding manager and employee relationships, high levels of trust in the leader can result in positive employee outcomes such as well-being [3,12], social integration, improved relationships and communication at work [4,12], job satisfaction [4,10,12], work-life balance [12], increased retention [13], better performance, organizational culture, and behavior [3], reduced competitive attitudes [4], adjustment, openness, organizational commitment [4], and work engagement [8,10]. Overall, high levels of trust in work environments and manager-employee relationships are associated with positive outcomes for both individuals and organizations. Conversely, low levels of trust can lead to negative consequences. Therefore, measuring trust accurately and effectively is crucial to better understand its impact and promote positive outcomes in the workplace.

The literature suggests that there is a limited selection of scales available to measure trust, a lack of replication in studies, weak evidence to support the construct validity of measurements, and limited consensus on the widespread usage of these instruments [14]. Indeed, the fragmentation in trust measurement can be attributed to the fact that trust is context-specific, meaning that it can vary depending on the situation, the individuals involved, and the environment in which the trust is being measured. As a result, it can be challenging to develop a universal instrument that accurately captures trust across all contexts, fields of study, and cultures [6,14]. Nonetheless, it is generally recommended that, where possible, previously validated measures should be adopted to measure trust rather than developing new measures [14]. This can help to increase the comparability of results across studies and facilitate cross-disciplinary research.

To our knowledge, the only scale available in Italian that measures trust is the Organizational Trust Inventory (long and reduced versions) [15]. A study [16] adopted the scale on a leader version and an organization version and used it on nurses and managers in hospital settings. The Organizational Trust Inventory aimed to measure trust at the organizational level, not at the work context level. Therefore, today, a scale that assesses trust in the workplace, with a focus on managers and employees, in Italian is lacking. One available instrument that responds to our needs is the Trust Me Scale [5]. This scale is designed to differentiate trust relationships between managers and employees within the specific work context and has presented good psychometric properties in different settings [5].

Adapting an instrument culturally to a different language from the original is important, especially when the instrument has been previously validated and has shown good psychometric properties. Translation and psychometric validation of instruments is essential for comparing results across studies and cultures and allows the collection of valid and reliable results in the new culture and advancing knowledge in the field [17]. Using psychometrically tested instruments also ensures that the results are robust and can be replicated in different settings, which is critical for building strong evidence, new theoretical perspectives, and informing clinical practice, education, and policy.

Research on the trust relationship between nurse managers and nurses in the workplace is limited internationally [4] and scarce in Italy. Further investigation of the concept of trust is crucial for building nursing science and advancing our understanding of human behavior in healthcare [1]. Understanding how trust is established, maintained, and nurtured can help healthcare professionals and leaders build more effective and successful relationships with their patients, colleagues, and followers. It can also strengthen healthcare organizations’ appeal to nursing professionals to attract, retain, and motivate the nursing workforce [13]. This is particularly important in the current context of a nursing shortage.

Therefore, the present study intends to contribute to the nursing literature on trust by testing the psychometric characteristics (validity and reliability) of the “Trust Me Scale” instrument in Italian among nurses and nurse coordinators and performing cultural validation.

## 2. Materials and Methods

### 2.1. Design

After a collaborative and iterative translation, we used a cross-sectional multi-center design [18].

### 2.2. Collaborative and Iterative Translation

The “Trust Me Scale” from its original language to Italian was translated and culturally adapted by following the methodological steps described by Douglas and Craig [18]. The translation was based on five basic stages involved in this process: pre-translation (establish equivalence), initial translation, pretesting, review, and administration.

In the pre-translation, the authors worked with a team of experts to establish the conceptual definition of the contents included in each item by considering a focus on category, functional, and construct equivalences. Category equivalence refers to the similarity of categories or labels used to describe phenomena across different cultures or languages. Functional equivalence refers to the extent to which a research instrument or measure functions similarly in different cultural groups. Construct equivalence refers to the similarity of the underlying meaning or concept being studied across different cultural groups or languages. This is often the most difficult type of equivalence to achieve, as some concepts may be unique to a particular culture or may be expressed differently in different languages and require psychometric testing. For instance, if the dimensionality of a scale is equal between different countries or cultures, this evidence could be considered an initial proof of construct equivalence, even if to directly assess this type of equivalence, multi-country studies are required to determine several levels of measurement invariances between countries. Therefore, achieving category equivalence and functional equivalence in questionnaire translation was the main aim of the pre-translation, even if the involved team kept in mind the need for construct equivalence. In this stage, the authors involved 11 experts: a psychologist who was an expert in translation and cultural equivalence, four nurse coordinators, and six staff nurses (all with previous experience in content validity studies).

After ensuring the equivalence of questions, an initial translation was conducted by an independent, parallel translation into the target language performed by a translator with experience in translations of self-report questionnaires. The pretesting involved the authors who did not take part in the translation of the scale. A review meeting with the translator, the experts, and the authors was held to decide on the final version and ensure accurate capture of the meaning of the items in the Italian language. The Italian version of the scale is available in Appendix A.

### 2.3. Sample and Setting of the Cross-Sectional Study

The study investigated the nature of trust relationships between nurse managers and registered nurses in various healthcare settings across different regions of Italy. For this reason, the work settings encompassed hospital units where nurses worked in groups, including wards, outpatient services, theatre rooms, intensive and semi-intensive care units, and community settings such as community care centers and public healthcare services, home care, and community homes.

To qualify for inclusion, registered nurses needed to be employed by a public or private healthcare organization, work collaboratively with other nurses or nurse assistants in a team, have a minimum of two months of experience in the service, and voluntarily agree to participate in the study. Conversely, exclusion criteria included being a freelance registered nurse, working in a solo setting without peer team collaboration, being new to the service for less than two months, not being assigned to a stable work setting, being back in the service for less than two months after a prolonged absence, and declining to participate in the study.

The sampling method employed was convenience-based, and participation was voluntary and anonymous. While power may not be the most relevant criterion for sample size determination in CFA, we chose to perform Monte Carlo simulations to estimate the required sample size for the analysis by considering the factor analysis results presented in the original study [5]. This approach was therefore preferred because we had two different subsamples (nurses and nurse coordinators), and we wanted to ensure that the desired power needed to reject the null hypothesis that ᴨ (the parameter) = 0 was met in both subsamples. The simulations were performed in Mplus version 8.1 (Los Angeles, CA: Muthén & Muthén) by employing 1000 replications (seed = 45,335; the residual variances of the factor indicators were 0.34; factor variances were fixed to one; factor correlation set to 0.65). A sample size of 180 or greater was needed to achieve a power ≥ 0.80, including 5% of missing data under the hypothesis of missing at random. This simulation helped us determine the minimum sample size required to achieve reasonably stable estimates of factor loadings.

### 2.4. Data Collection

The principal investigators (DI & DT) promoted the study through the Italian Scientific Society for the Direction and Management of Nursing (SIDMI) network. They met with Nurse Executive Officers who expressed interest and enlisted local contact persons to explain the study to nurse managers and nurses at each study center and encourage participation. The survey link was distributed via institutional email addresses, and regular communication occurred between the local contacts and principal investigators to share participation feedback.

Data were collected between August 2022 and January 2023 using an online survey on the Google Forms platform. The survey detailed the study’s objectives and participation process, followed by the informed consent and data treatment section. Nurses were given the option to participate in the study and complete the entire survey or only certain parts of it. All data were collected without identifying information to ensure anonymity.

### 2.5. Measurements

The survey began by collecting socio-demographic information, including age (years), sex (male, female, other), the highest level of education, overall work experience (years), and work experience in the last service/ward (years).

In addition, nurses were asked about their intention to leave the current work setting, the organization, or the profession altogether. This intention was measured using single items with binary response options: 1 (yes, I intend to leave the service within the next six months) and 2 (no, I do not intend to leave the service). Literature suggests using single-item measurements when evaluating a concrete construct, such as in this case [19].

Nurses were asked to express their satisfaction levels with their role, multidisciplinary work, leader, and organization. We used single items with a 5-point Likert scale ranging from 0 (very unsatisfied) to 4 (very satisfied) to collect this information [20]. A higher score on the Likert scale indicates a higher level of satisfaction.

To measure organizational commitment, we used the Organizational Commitment Scale from the Questionnaire on Experience and Work Evaluation (QEEW 2.0 © SKB) developed by van Veldhoven et al. [21]. The scale comprises six positively-worded items and is rated on a 5-point Likert scale ranging from 0 (strongly agree) to 4 (strongly disagree). The scoring of the scale ranges from 0 to 100, with lower scores indicating higher levels of organizational commitment. The scale has solid psychometric properties, with an internal consistency of 0.80, and is already available in Italian.

We used the Trust Me Scale developed by Tzafrir et al. [5] to measure trust. It was translated and adapted as previously described using a collaborative and iterative translation, following the methodological steps described by Douglas and Craig [18]. The scale measures trust as a multidimensional construct comprising 16 positively-worded items. The items are rated using a 5-point Likert scale ranging from 1 (strongly disagree) to 5 (strongly agree), with three items reversed to measure mistrust. The score is calculated as the mean, with higher scores indicating greater trust. The scale has undergone rigorous psychometric validation and identified three factors. The first factor measures harmony, which reflects a feeling of belonging and mutual support in manager-employee relationships within the context of work. The second factor measures reliability, which indicates the consistency and adherence to established processes in the relationships among leaders and members at work. The third-factor measures shared concern for the well-being of others, which is weighed against personal interests. The factors presented a Cronbach’s alpha of 0.85 for harmony, 0.87 for reliability, and 0.80 for concern.

### 2.6. Ethical Considerations

The research received approval from the local Ethics Committee and was approved by the Board of Directors of each participating center. The study was carried out in accordance with ethical standards and the principles outlined in the Helsinki Declaration [22]. Before participating in the study, all participants were given information about the research and signed an online informed consent form. Data access was restricted solely to the research team.

### 2.7. Statistical Analysis

The study initially presented an overview of the characteristics of nurses and nurse coordinators through the utilization of descriptive statistics and inferential comparisons with a significance level adjusted according to Bonferroni (adjusted α = 0.003). This enabled the identification of the distribution of respondents’ traits. Following the Trust Me Scale translation, psychometric validation was conducted using four main steps [18].

In the first step, two fully saturated models were constructed, one for each subgroup (nurses and nurse managers), whereby all factor loadings were freely estimated, and no restrictions were placed on the residual variances. Despite using a confirmatory framework, these two fully saturated models were developed for each subgroup to describe the behavior of the factor loadings and residual variances. This approach was adopted considering that the authors indicated the potentially problematic nature of the negatively worded items in the original work due to the possible misunderstanding of their meanings [5]. In this step, items that had statistically significant factor loadings lower than 0.40 (i.e., testing the null hypothesis that loading is at least 0.40 while rejecting the test would lead to item removal), indicating less than 16% of common variances with their respective factors, were considered for removal from the scale. Factor loadings were reported along with their standard errors to provide the precision of the estimates. In addition, factor loadings were standardized for comparability by designating one indicator for each factor as the reference point and fixing its factor loading at 1, then estimating the factor loadings of the other indicators relative to this reference point. Modification indices (MI) in Mplus were also used to identify areas of the models where fit could be improved by allowing additional parameters to be estimated. The indices indicated the extent to which the model’s fit could be improved by adding a particular parameter to the model. MI can also detect unmodelled residual correlations and suggest adding or removing particular paths in the model. More precisely, strong correlations of residual variables might suggest that the model did not fully capture the covariance between those items by indicating unmodelled residual correlations. MI can indicate which residual variables are contributing to this over-correlatedness and suggest the modification of the model to improve its fit. In the case of this scale, if negatively worded items caused correlations of the residuals significantly different than expected based on theoretical considerations or previous research [5], the authors had the a priori strategy in accordance with the authors that developed the scale to delete these items instead of constraining the model, acknowledging that these items (items 5, 11, and 13) were indicated a possible source of misunderstanding by previous research [5]. In other words, while it was possible to account for residual correlations by including scale reliability residual variables in the model, we chose to follow an a priori strategy of deleting items with correlations between residual variables different than expected based on theoretical expectations derived from previous research [5]. This decision was informed by concerns raised by experts involved in the collaborative and iterative translation process about the understandability of these items. These concerns were also shared with the authors of the original scale, who suggested considering the removal of items that might lead to misinterpretations. In this research, expected values for correlations between residual variables were based on previous research findings and the theoretical framework guiding the development of the Trust Me Scale [5] with expected correlations lower than 0.6. In addition, residual variances higher than 0.6 were defined as extreme residual variances in this research.

In the second step, the scale without problematic items was tested by employing CFA models in the two subgroups, and the fit indices of the models resulting from the second step were compared with the ones resulting from the first step, showing a delta (Δ) indicating whether the fit in explaining sample statistics improved after having removed potentially problematic items. The employed fit indices were the Comparative Fit Index (CFI), Turker and Lewis Index (TLI), Root Mean Square Error of Approximation (RMSEA), and Standardised Root Mean Square Residual (SRMR) [23]. CFI and TLI are two fit indices that range from 0 to 1, with values closer to 1 indicating a better fit: values between 0.90 and 0.95 (or higher) are considered good, while values lower than 0.90 show a poorer fit. RMSEA reflects the extent to which the model’s predicted values deviate from the observed values, taking into account the complexity of the model. RMSEA values range from 0 to 1, with values below 0.05 indicating a well-fitting model, values between 0.05 and 0.08 indicating a moderate fit, and values greater than or equal to 0.10 indicating a poor fit. RMSEA values with 90% confidence intervals ranging from 0.05 to 0.08 indicate a good fit. SRMR measures the difference between the observed and predicted covariances. SRMR values range from 0 to 1, with values less than or equal to 0.08, indicating a good fit for the model of interest.

In the third step, a multiple-indicator multiple-cause (MIMIC) model was used to test the measurement invariance of the Trust Me Scale between nurses and nurse managers, and it was performed in the overall model. Even if measurement invariance is not a prerequisite for comparing factors in different groups, it could be a desirable psychometric property. Therefore, a MIMIC model can test the measurement invariance by simultaneously estimating the relationships between the latent variables and the observed variables while also accounting for the effects of the group variable (nurses vs. nurse managers) on the observed variables. This approach allowed for testing the equivalence of the factor structure across groups while also controlling for potential group differences in the measurement of the construct. In this step, McDonald’s ω coefficients were employed to determine the scale reliability of the domains.

In the final step, we conducted hypothesis testing to assess the construct validity of the Trust Me Scale. Following previous studies, our a priori hypotheses were that participants who had no intention to leave their ward/service, the company/hospital, and the nursing profession would score higher on the Trust Me Scale domains [1,3,4,7,9,10,11,24,25]. We tested these hypotheses using point-biserial rho correlations. Additionally, we hypothesized that participants who reported higher satisfaction with their current role, multidisciplinary work, and leadership and higher organizational commitment would score higher on the Trust Me Scale domains [7,9,11]. We used Pearson’s rho correlations to test these hypotheses. We expected all correlations to be positive, except for those concerning organizational commitment, where lower scores indicated higher commitment.

Analyses were performed using Stata Statistical Software: Release 17 (StataCorp. 2021; College Station, TX, USA: StataCorp LLC.) and Mplus version 8.1 (Los Angeles, CA, USA: Muthén & Muthén) for the CFA models that were performed with robust maximum likelihood estimator.

## 3. Results

### 3.1. Sample

The study sample comprised 683 nurses and 188 nurse coordinators; their characteristics are presented in Table 1. The majority of the nurses were recruited from northern Italian regions (n = 337; 49.3%), whereas most nurse managers were from southern Italian regions (n = 90; 47.9%) (*p* < 0.001). In both groups, most respondents were female and employed in public hospitals (rates higher than 70% in both groups and for both variables), with no significant difference between the groups (*p* = 0.193). The mean age and work experience of nurse managers (52.35 ± 6.65 years and 30.42 ± 8.69 years, respectively) were significantly higher than those of nurses (42.57 ± 11.25 years and 16.94 ± 11.14 years, respectively) (*p* < 0.001). Nurse managers also showed higher rates of postgraduate education, including master of science (*p* < 0.001). Rates of intention to leave the ward/service and the company/hospital tended to be higher among nurses compared to nurse managers: 26.8% vs. 16.5% (*p* = 0.004) and 22.7% vs. 15.4% (*p* = 0.031), respectively. However, the intention to leave the profession was similarly distributed among nurses (17.6%) and nurse managers (12.2%) (*p* = 0.080). Median satisfaction scores for the current role were equivalent in both groups (*p* = 0.138), while scores for multidisciplinary work and company/hospital were significantly higher among nurses (*p* < 0.001). Nurses also reported higher satisfaction regarding leadership compared to nurse managers (*p* < 0.001). Additionally, nurse managers demonstrated higher levels of organizational commitment (median = 37.5, IQR = 33–62.5) compared to nurses (median = 50, IQR = 33–62.5) (*p* < 0.001).

Descriptive statistics of the Trust Me Scale items are shown in Figure 1 (English wording), while the Italian-translated items are available in Appendix A.

### 3.2. Validity Testing

#### 3.2.1. Step 1: Fully Saturated Models

The sample size for splitting the sample into two groups (nurses and nurse managers) was adequate and ensured a power higher than 80% in both models, following the Monte Carlo simulation performed after the scale translation. 

In the subgroup of nurses, the posited model with three factors showed a minimally adequate fit to sample statistics: χ^2^ (101, N = 683) = 555.886, *p* < 0.001; CFI = 0.911; TLI = 0.897; RMSEA = 0.081 (90% CI = 0.075–0.088); SRMR = 0.052. The factor loadings are shown in Table 2 and ranged from 0.287 (item 5) to 0.895 (item 7). Residual variances ranged from a maximum of 0.918 (standard error, SE = 0.024; item 5) to 0.200 (SE = 0.024; item 7). The correlation among the residual variables of item 13 and item 11 was high (MI = 173.942), and, as expected, the model did not fully capture the covariance between those items.

In the subgroup of nurse managers, the posited model with three factors showed a poorly adequate fit to sample statistics: χ^2^ (101, N = 188) = 183.273, *p* < 0.001; CFI = 0.897; TLI = 0.877; RMSEA = 0.066 (90% CI = 0.051–0.081); SRMR = 0.058. The factor loadings are shown in Table 2 and ranged from 0.187 (item 5) to 0.729 (item 4). Residual variances ranged from a maximum of 0.965 (SE = 0.030; item 5) to 0.460 (SE = 0.085; item 12). The correlation among the residual variables of item 13 and item 11 was high (MI = 163.921), and as expected, the model did not fully capture the covariance between those items.

Overall, Table 3 shows the model fit of the two models. In this step, the authors decided to delete the items that did not behave consistently with what the model explained: item 5 had a poor factor loading and an extreme residual variance equal to 0.856, while items 11 and 13 had extreme residual variances (respectively, 0.610 and 0.658), where residual variables were highly inter-correlated and not captured by the unconstrained model.

#### 3.2.2. Step 2: CFA Models after Having Deleted Items 5, 11, and 13

In the subgroup of nurses, the posited model with three factors showed a good fit to sample statistics: χ^2^ (62, N = 683) = 197.567, *p* < 0.001; CFI = 0.968; TLI = 0.956; RMSEA = 0.057 (90% CI = 0.048–0.066); SRMR = 0.025. The factor loadings are shown in Table 2 and ranged from 0.503 (item 10) to 0.895 (item 7). Residual variances ranged from a maximum of 0.607 (SE =0.045; item 14) to 0.174 (SE = 0.015; item 6). The posited model well explained the correlations among the residual variables.

In the subgroup of nurse managers, the posited model with three factors showed a good fit to sample statistics: χ^2^ (62, N = 188) = 98.278, *p* < 0.001; CFI = 0.946; TLI = 0.932; RMSEA = 0.056 (90% CI = 0.034–0.076); SRMR = 0.043. The factor loadings are shown in Table 2 and ranged from 0.414 (item 14) to 0.760 (item 4). Residual variances ranged from a maximum of 0.585 (SE = 0.074; item 14) to 0.159 (SE = 0.077; item 6). The posited model well explained the correlations among the residual variables.

The differences between the fit indices of step 2 and step 1 are shown in Table 3.

#### 3.2.3. Step 3: MIMIC Model in the Overall Sample

The posited model with three factors showed a good fit to sample statistics: χ^2^ (72, N = 188) = 314.02 *p* < 0.001; CFI = 0.936; TLI = 0.916; RMSEA = 0.074 (90% CI = 0.064–0.080); SRMR = 0.039. The factor loadings are shown in Table 2 and ranged from 0.516 (item 10) to 0.870 (item 7). Residual variances ranged from a maximum of 0.603 (SE = 0.039; item 14) to 0.169 (SE = 0.021; item 6). The correlations among the residual variables were well explained by the posited model. In addition, the nominal covariate of the model labeled as “group” (1 = nurses; 2 = nurse managers) had no linear relationships with the factors (all *p*-values were higher than 0.05), indicating that the dimensionality was equal in the two groups.

McDonald’s ω coefficients were adequate for each domain: harmony, reliability, and concern had McDonald’s ω coefficients equal to 0.863, 0.856, and 0.886, respectively.

#### 3.2.4. Step 4: Hypothesis Testing

The priory hypotheses were tested using correlational analyses, which are shown in Table 4, and all the hypotheses were confirmed. Positive correlations were detected between the scale domains with the intention to leave variables and satisfaction levels, indicating higher trust in participants with no intention to leave. Negative correlations were shown between scale domains and organizational commitment, indicating higher trust scores in participants with higher organizational commitment.

## 4. Discussion

Testing the psychometric characteristics of the Trust Me Scale in its Italian version is crucial to validate its use in the Italian context, specifically among nurses and nurse managers. Translating a scale from one language to another is not always straightforward, and the cultural differences between countries can impact how individuals perceive and respond to the items [18]. Therefore, after the scale translation following precise methodological steps [18], assessing the reliability and validity of its Italian version is essential to ensure that the results obtained from its application in future utilization are trustworthy and accurate. Furthermore, a validated Italian version of the scale could provide researchers and healthcare professionals with a valuable tool to measure trust in healthcare providers, which is a critical component of a healthy organizational culture. A healthy organizational culture in healthcare is essential for providing safe and effective care to patients, and trust among healthcare providers is a critical component of such a culture [26,27]. In this regard, the Trust Me Scale can help identify areas for improvement in organizational culture, which can ultimately lead to better patient outcomes and contribute to public health.

The main results of this study are related to the information regarding the validity and reliability of the Italian version of the Trust Me Scale, which encompasses 13 items measuring the domains described in the original scale [5]. The approach employed to validate the scale was a rigorous and systematic psychometric validation process, which enhanced the reliability and validity of the scale in its Italian version [28]. In the first step, fully saturated models were constructed to describe the behavior of factor loadings and residual variances to evaluate items that might be problematic, as previously described [5]. In the second step, items 5, 11, and 13 were removed from the scale, and CFA models were employed to test the fit indices in the two subgroups (nurses and nurse managers). In the third step, a MIMIC model was used to test the measurement invariance of the Trust Me Scale between nurses and nurse managers, and McDonald’s ω coefficients were employed to determine the scale reliability of the domains. Finally, hypothesis testing was conducted to assess the construct validity of the Trust Me Scale.

The deleted items in the first step were the negatively worded situations (items 5, 11, and 13). Negative wording can make it harder for participants to understand the question, which can lead to misunderstandings and errors in responses [29]. In the traditional perspective for developing self-report measures, using both positively and negatively worded items in self-report measures is a common practice to reduce response bias [30]. This approach assumes that both types of items measure the same construct and can minimize acquiescent responding. However, negatively worded items may not be fully equivalent to their positively worded counterparts, and respondents may answer them differently due to the wording effect [31]. While negatively worded items are included in self-report scales to reduce acquiescence bias (i.e., participants tend to agree with items regardless of their actual beliefs or experiences), they can introduce response biases if respondents find them difficult to understand. This is because participants may be more likely to agree with the item even if it does not accurately reflect their feelings or experiences (i.e., acquiescence bias induced by the need to focus on the item more than expected by respondents). As a result, scores on the scale can be inflated, potentially compromising the validity and reliability of the results. For these reasons, after the first step of the analytical process, the authors deleted the negatively worded items that did not show fit with the unconstrained CFA model, which was employed to test the original dimensionality.

The employed hypothesis testing confirmed that participants who had no intention to leave their ward/service, the company/hospital, and the nursing profession would score higher on the Trust Me Scale domains [1,3,4,7,9,10,11,24,25], as well as those with higher organizational commitment [7,9,11]. This study found that nurses and nurse managers who trust their organization are more likely to develop stronger organizational commitment, which in turn makes them less likely to have the intention to leave. Therefore, trust plays a significant role in shaping nurses’ commitment to their organization and their decision to stay or leave the ward, the organization, or the profession. The study results provide evidence that the measured domains of trust align with the theoretical expectations and are related to the intention to leave, job satisfaction, and organizational commitment. These results support the construct validity of the Italian version of the Trust Me Scale, highlighting its relevance in understanding the complex relationships between trust, commitment, and turnover intentions among healthcare professionals.

Regarding the reliability for each domain, the generalizability interpretation of coefficient alpha is usually favored over other interpretations unless the true scores can be accurately estimated by a latent variable model that has been well-validated and is sufficiently restrictive to provide a reliable estimate of reliability, like McDonald’s ω coefficient. In this study, differently than the previously published original study for developing and validating the Trust Me Scale [5], the true score (i.e., scale) reliability was assessed using McDonald’s ω coefficients because they provide a more restrictive estimate of the reliability [32]. While Cronbach’s alpha is based on classical test theory, McDonald’s ω coefficient is based on the factor model (restrictive approach) and assumes that items can have different factor loadings and may measure different aspects of the construct [33]. This characteristic makes McDonald’s ω coefficients particularly useful for exploring the scale reliability of the Trust Me Scale in this initial stage of Italian validation. In this study, the three domains of the scale showed adequate internal consistency.

The results showing the validity and reliability of the Trust Me Scale in its Italian version suggest that the scale can be used to measure trust among healthcare professionals, which can have important implications for clinical practice and research. Measuring trust in clinical practice can help healthcare professionals understand and manage their relationships with patients and colleagues. For example, measuring trust between colleagues in the nurse and leader relationships can help identify potential areas of conflict or tension in the workplace. These assessments work to improve collaboration and teamwork [27]. In research, the Trust Me Scale can measure trust in different healthcare contexts and explore the relationship between trust and outcomes such as patient satisfaction and healthcare provider job satisfaction. This utilization can help to identify factors that contribute to trust in healthcare settings and to develop interventions to improve trust and healthcare outcomes.

This study has several limitations that require to be acknowledged. Firstly, the study only collected data at one point in time (cross-sectional), which limits the ability to test the stability of the Trust Me Scale over time. It is unclear how stable the measure is over time or how it may change in response to different interventions or changes in the healthcare context. Secondly, the study was conducted with a convenience sampling procedure. Therefore, the external validity of the findings to other healthcare settings, populations, and cultures may be limited. Thirdly, the study relied solely on self-report measures, which may be subject to response biases or social desirability effects. Conversely, the main strengths of the study are given by the relatively large sample of nurses and nurse managers, which increases the generalizability of the findings to other Italian healthcare settings; furthermore, the authors used rigorous statistical methods, including CFA and MIMIC modeling, to establish the validity and reliability of the Trust Me Scale. Considering the implication of this research, it is relevant to highlight that this study adds to the literature on trust in healthcare by providing a validated instrument for measuring trust in the Italian context among nurses and nurse coordinators, and the validated scale can be used in future research to explore the relationships between trust and other variables, such as patient outcomes and healthcare provider behaviors.

## 5. Conclusions

The validation of the Trust Me Scale in its Italian version is crucial for measuring trust in nurses and nurse managers. The study’s rigorous psychometric validation process enhanced the reliability and validity of the Italian version of the scale. Removing negatively worded items improved the clarity of the questions and prevented response bias. This validated Italian version of the Trust Me Scale can help identify areas for improvement in organizational culture, which is essential for providing safe and effective care to patients. Trust among healthcare providers is a critical component of a healthy organizational culture and can ultimately lead to better patient outcomes, contributing to public health. Future research can address these limitations to provide more robust evidence on the scale’s validity and reliability in different healthcare settings.

## Figures and Tables

**Figure 1 healthcare-11-01086-f001:**
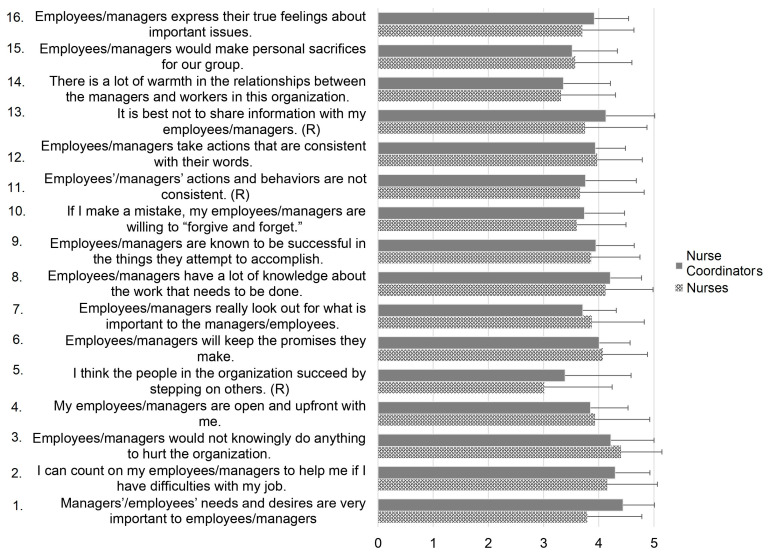
Bargraph of the items 1–16.

**Table 1 healthcare-11-01086-t001:** Sample characteristics (nurses and nurse managers).

		Nurses (N = 683)		Nurse Managers (N = 188)		Comparisons
		N	%	N	%	
Region						
	Northern regions	337	49.3	71	37.8	χ^2^ (2, N = 871) = 18.47, *p* < 0.001
	Central regions	134	19.3	27	14.4	
	Southern regions	212	31	90	47.9	
Setting						
	Public hospital	479	70.1	141	75	χ^2^ (2, N = 871) = 3.29, *p* = 0.193
	Comunity care service	91	13.3	26	13.8	
	Private hospital delivering public service	113	16.5	21	11.2	
Age						
	Years (mean; SD)	42.57	11.25	52.35	6.65	t_(510.057)_ = −15.06, *p* < 0.001
Sex						
	Female	510	75	144	76.6	χ^2^ (1, N = 871) = 0.782, *p* = 0.676
Work experience						
	Years (mean; SD)	16.94	11.14	30.42	8.69	t_(188.41)_ = −5.96, *p* < 0.001
Work experience in the last ward/service						
	Years (mean; SD)	8.32	8.3	8.85	9.12	t_(188.35)_ = −1.09, *p* = 0.277
Educational background						
	BSc or equivalent title	181	26.5	10	5.4	χ^2^ (5, N = 871) = 260.821, *p* < 0.001
	Postgraduate certificate after BSc	286	41.9	112	59.6	
	Master of Science	162	23.7	37	19.7	
	Postgraduate certificate after MSc	44	6.4	28	14.9	
	Other postgraduate education	1	0.1	0	0	
	PhD	9	1.3	1	0.5	
Intention to leave the ward/service						
	Yes	183	26.8	31	16.5	χ^2^ (1, N = 871) = 8.45, *p* = 0.004
Intention to leave the company/hospital						
	Yes	155	22.7	29	15.4	χ^2^ (1, N = 871) = 4.67, *p* = 0.031
Intention to leave the nursing profession						
	Yes	120	17.6	23	12.2	χ^2^ (1, N = 871) = 3.06, *p* = 0.080
Satisfaction regarding the current role						
	Score [0 = completely not satisfied; 4 = completely satisfied] (median; IQR)	3	2–3	3	2–4	U = 60,010.5 (z = −1.18), *p* = 0.138
Satisfaction regarding multidisciplinary work						
	Score [0 = completely not satisfied; 4 = completely satisfied] (median; IQR)	3	2–3	3	3–3	U = 53,993 (z = −3.63), *p* < 0.001
Satisfaction regarding the leadership						
	Score [0 = completely not satisfied; 4 = completely satisfied] (median; IQR)	3	2–4	3	3–3	U = 51,562 (z = −4.53), *p* < 0.001
Satisfaction with the company/hospital						
	Score [0 = completely not satisfied; 4 = completely satisfied] (median; IQR)	3	2–3	3	2–3	U = 54,436.5 (z = −3.38), *p* < 0.001
Organizational Commitment Scale						
	Score [0 = highest commitment; 100 = lowest commitment] (median; IQR)	50	33–62.5	37.5	25–50	U = 47,659.5 (z = −5.43), *p* < 0.001

*Legend*: SD = standard deviation; BSc = Bachelor of Sciences (in Nursing); MSc = Master of Sciences (in Nursing); IQR = interquartile range. Statistically significant differences require *p* < adjusted α (0.003).

**Table 2 healthcare-11-01086-t002:** Factor loadings in step 1 (without deletion) and step 2 (after item deletion) in the subgroups of nurses and nurse coordinators, and in step 3 (overall sample analyzed with MIMIC model).

	Step 1		Step 2		Step 3: MIMIC
	Nurses	Nurse Coordinators	Nurses	Nurse Coordinators	Overall
Factor 1: Harmony					
Item 8	0.809 (0.017)	0.669 (0.048)	0.809 (0.017)	0.663 (0.048)	0.796 (0.030)
Item 9	0.780 (0.025)	0.735 (0.044)	0.781 (0.025)	0.734 (0.043)	0.773 (0.031)
Item 14	0.618 (0.032)	0.408 (0.076)	0.619 (0.032)	0.414 (0.076)	0.590 (0.034)
Item 15	0.806 (0.019)	0.662 (0.051)	0.806 (0.019)	0.666 (0.050)	0.781 (0.031)
Item 16	0.815 (0.018)	0.606 (0.053)	0.814 (0.018)	0.609 (0.052)	0.791(0.029)
Factor 2: Reliability					
Item 1	0.797 (0.022)	0.498 (0.058)	0.797 (0.022)	0.495 (0.057)	0.734 (0.035)
Item 6	0.855 (0.014)	0.702 (0.123)	0.858 (0.014)	0.698 (0.125)	0.842 (0.026)
Item 7	0.895 (0.013)	0.674 (0.055)	0.895 (0.014)	0.678 (0.056)	0.870 (0.027)
Item 11	0.551 (0.041)	0.478 (0.084)	–	–	–
Item 12	0.832 (0.027)	0.729 (0.058)	0.831 (0.027)	0.715 (0.060)	0.821 (0.029)
Factor 3: Concern					
Item 2	0.802 (0.023)	0.701 (0.048)	0.803 (0.023)	0.699 (0.047)	0.788 (0.032)
Item 3	0.584 (0.034)	0.475 (0.076)	0.581 (0.034)	0.469 (0.075)	0.550 (0.029)
Item 4	0.850 (0.015)	0.761 (0.040)	0.848 (0.015)	0.760 (0.041)	0.837 (0.030)
Item 5	0.287 (0.041)	0.187 (0.081)	–	–	–
Item 10	0.492 (0.040)	0.652 (0.050)	0.503 (0.039)	0.650 (0.051)	0.516 (0.035)
Item 13	0.486 (0.045)	0.234 (0.108)	–	–	–
Correlations					
Harmony–Reliability	0.927 ***	0.901 ***	0.901 ***	0.905 ***	0.916 ***
Harmony–Concerns	0.917 ***	0.913 ***	0.906 ***	0.920 ***	0.908 ***
Reliability–Concerns	0.925 ***	0.932 ***	0.929 ***	0.937 ***	0.924 ***

*Legend*: MIMIC = multiple-cause model; *** indicates *p* < 0.001. *Note*: Numbers in the table present fully standardized factor loading with standard error on parenthesis for the factors; for correlations numbers present Pearson’s rho test.

**Table 3 healthcare-11-01086-t003:** Fit information of the performed models.

	Step 1		Step 2		Δ (Step 2–Step 1)		MIMIC
	Nurses	Nurse Managers	Nurses	Nurse Managers	Nurses	Nurse Managers	Overall
χ^2^	555.886 (DF = 101)	183.273 (DF = 101)	197.567 (DF = 62)	98.278 (DF = 62)	−358.319	−84.995	314.02 (DF = 72)
CFI	0.911	0.897	0.968	0.946	0.057	0.049	0.936
TLI	0.895	0.877	0.956	0.932	0.061	0.055	0.916
RMSEA (90%CI)	0.081 (0.075–0.088)	0.066 (0.051–0.081)	0.057 (0.048–0.066)	0.056 (0.034–0.076)	−0.024	−0.01	0.074 (0.064–0.080)
SRMR	0.052	0.058	0.025	0.043	−0.027	−0.015	0.039

*Legend*: CFI = comparative fit index; TLI = Tucker–Lewis index; RMSEA = Root Mean Square Error Of Approximation; CI = confidence interval; SRMR = Standardized Root Mean Square Residual; DF = degrees of freedom.

**Table 4 healthcare-11-01086-t004:** Correlations between the Trust Me Scale and intention to leave, satisfaction levels, and organizational commitment.

	Trust Me Scale			
	Harmony	Reliability	Concern	Total Score
Intention to leave the ward/service	0.151 **	0.163 **	0.173 **	0.186 **
Intention to leave the company/hospital	0.262 **	0.229 **	0.202 **	0.263 **
Intention to leave the nursing profession	0.175 **	0.138 **	0.109 **	0.169 **
Satisfaction regarding the current role	0.274 **	0.267 **	0.275 **	0.315 **
Satisfaction regarding multidisciplinary work	0.345 **	0.337 **	0.239 **	0.376 **
Satisfaction regarding the leadership	0.464 **	0.489 **	0.441 **	0.512 **
Satisfaction with the company/hospital	0.325 **	0.275 **	0.258 **	0.322 **
Organizational Commitment Scale	−0.421 **	−0.317 **	−0.270 **	−0.378 **
Harmony	–	0.833 **	0.758 **	0.906 **
Reliability	–	–	0.815 **	0.918 **
Concern	–	–	–	0.882 **

*Legend*: ** indicates *p* < 0.01.

## Data Availability

The data presented in this study are available on reasonable request from the corresponding author.

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
