# Peer review of "Cultural Adaptation and Psychometric Properties of the Trust Me Scale—Italian Version: A Validation Study"

_healthcare, 2023, doi:10.3390/healthcare11081086_

Round 1

Reviewer 1 Report

You did a fine validation study that respects all psychometric procedures.   I can see the usefulness of your validation in using it in the health sector in Italy

Reviewer 2 Report

This is a well executed validation study. It is careful in presenting how the authors carried out the validation. The translations and the statistical analysis are appropriate.

My only suggestion is that the author include a Table around line 374 that notes the items that were asked. It would be useful to provide 3 columns: item number; english version of the item; italian version of the item. While this information is available in the supporting information, it would save me from having to go back and forth. It would certainly make Table 2 much easier to follow.

Reviewer 3 Report

The submitted manuscript analyzes an Italian translation of the trust me scale questionnaire regarding their psychometric properties and studies some external validity criteria. I found the manuscript well-structured. I have a few comments:
1.    Sec. 2.1., l. 172: Please also provide an English translation of the questionnaire in the appendix.
2.    L. 199: I am unsure whether power is the relevant criterion for sample size determination in confirmatory factor analysis (CFA). The authors perform multiple steps of model refinement. Hence, item loadings and residual correlations should have sufficiently small standard errors in order to protect against model modifications based on statistical randomness. See my next comment.
3.    Sect. 2.6.: Items were removed if their loadings were below 0.33. The issue is that for a small n of 188 in the group of nurse managers, statistical significance should be taken into account. I would test whether an item loading is statistically significantly smaller than 0.40 (i.e., testing the null hypothesis that a loading is at least 0.40 while rejecting the test would lead to item removal). Otherwise, you change an instrument just according to statistical noise.
4.    L. 280 and other places in the manuscript: It is inappropriate to speak of “over-correlated residuals variances”. First, there only can be residual correlations, that is, correlations of residual variables. Residual variances are parameters. Second, the term “over-correlated” is ill-defined. One should only speak of unmodelled residual correlations.
5.    Throughout the text: Define “extreme residual variances”.
6.    Connected to 4.: Residual correlations could be simply accommodated in CFAs by including correlations among residual variables. There is no need to remove the items from the analysis. It would prefer not to exclude both items but to retain at least one of the items (maybe the positively worded one).
7.    Table 2: Include standard errors for all parameters.
8.    L. 295: It is preferable to say that there was no association of gender with the factors.
9.    L. 307: I would like to note that measurement invariance might be a desirable psychometric property, but it is no prerequisite for comparing factors in the two groups.
10.    Table 4: Include standard errors.
11.    Table 2, Table 4: Report correlations of the latent factors in the two groups.
12.    Table 2: Authors report factor loadings for the two groups in Step 1 and Step 2 as well as the joint MIMIC model in Step 3. It is obvious that loadings will be confounded with true factor variances. To enable comparability, either use the output of the alignment procedure to report factor loadings located on a comparable scale or define a reference indicator for each scale with a factor loading of 1.
13.    L. 399: typo “and 0.886”
14.    L. 468: The authors refer to a reference of ordinal alpha. However, it seems that they use the robust maximum likelihood estimator and do not apply an ordinal CFA. Please explain.
15.    L. 470: It is illegitimate to say that McDonald’s omega is preferable to Cronbach‘s alpha. It is wrong that alpha relies on a factor model for items with equal loadings. Alpha is based on item sampling assuming exchangeable items (see Cronbach & Shavelson, 2004; Meyer, 2010; Ellis, 2021).

References:
Cronbach, L.J. and Shavelson, R.J., 2004. My current thoughts on coefficient alpha and successor procedures. Educational and Psychological Measurement, 64(3), pp.391-418.

Ellis, J.L., 2021. A test can have multiple reliabilities. Psychometrika, 86(4), pp.869-876.

Meyer, J. P. (2010). Understanding measurement: Reliability. Oxford University Press. https://global.oup.com/academic/product/understanding-measurement-reliability-9780195380361?cc=de&lang=en&

Round 2

Reviewer 3 Report

I have a few remaining comments regarding the revised version of the manuscript:
1.    Abstract, line 31: “items 11 and 13 were removed following an a priori strategy focused on deleting items with residual variances that were significantly different than expected based on theoretical expectations derived from previous research.”. I do not think that “residual variances” turned out to be different compared to expectations. Instead, correlations between residual variables (also referred to as residual correlations) were empirically observed. The authors should change this phrase in the whole manuscript. The same issue also occurs in l. 299, 303, 306, 386, 393, 408, etc. This issue should be resolved before publication.
2.    L. 500: Authors use the phrase “internal consistency reliability” and argue that \omega assesses this kind of reliability. Omega is based on a factor model and should not be confounded with internal consistency. The latter term should be reserved for Cronbach’s Alpha. Consider replacing “internal consistency reliability” with “scale reliability” or “true score reliability”.
